



# EuLerian Identification of ascending Air Streams (ELIAS 2.0) in Numerical Weather Prediction and Climate Models. Part I: Development of deep learning model

Julian F. Quinting[1] and Christian M. Grams[1]

[1]Institute of Meteorology and Climate Research (IMK-TRO), Karlsruhe Institute of Technology, Germany

**Correspondence:** Julian F. Quinting (julian.quinting@kit.edu)

**Abstract.** Physical processes on the synoptic scale are important modulators of the large-scale extratropical circulation. In particular, rapidly ascending air streams in extratropical cyclones, so-called warm conveyor belts (WCBs), modulate the upper-tropospheric Rossby wave pattern and are sources and magnifiers of forecast uncertainty. Thus, from a process-oriented perspective, numerical weather prediction (NWP) and climate models should adequately represent WCBs. The identification of
WCBs usually involves Lagrangian air parcel trajectories that ascend from the lower to the upper troposphere within two days. This requires numerical data with high spatial and temporal resolution which is often not available from standard output and requires expensive computations. This study introduces a novel framework that aims to predict the footprints of the WCB inflow, ascent, and outflow stages over the Northern Hemisphere from instantaneous gridded fields using convolutional neural networks (CNNs). With its comparably low computational costs and relying on standard model output alone the new diagnos-
tic enables the systematic investigation of WCBs in large data sets such as ensemble reforecast or climate model projections which are mostly not suited for trajectory calculations. Building on the insights from a logistic regression approach of a previous study, the CNNs are trained using a combination of meteorological parameters as predictors and trajectory-based WCB footprints as predictands. Validation of the networks against the trajectory-based data set confirms that the CNN models reliably replicate the climatological frequency of WCBs as well as their footprints at instantaneous time steps. The CNN models
significantly outperform previously developed logistic regression models. Including time-lagged information on the occurrence of WCB ascent as a predictor for the inflow and outflow stages further improves the models' skill considerably. A companion study demonstrates versatile applications of the CNNs in different data sets including the verification of WCBs in ensemble forecasts. Overall, the diagnostic demonstrates how deep learning methods may be used to investigate the representation of weather systems and of their related processes in NWP and climate models in order to shed light on forecast uncertainty and
systematic biases from a process-oriented perspective.

## 1 Introduction

Warm conveyor belts (WCBs; e.g., Carlson, 1980) are coherent, cross-isentropically ascending air streams in extratropical cyclones. The WCB air masses originate from the boundary layer in the warm sector of extratropical cyclones (WCB inflow), ascend across the cyclones' warm front (WCB ascent), and reach the upper troposphere (WCB outflow) within two days.





Numerous studies emphasize that WCBs have a major effect on the dynamics (e.g., Wernli and Davies, 1997; Pomroy and
Thorpe, 2000; Grams et al., 2011; Binder et al., 2016; Bosart et al., 2017) and forecast skill and predictability of the large-scale
midlatitude flow (e.g., Lamberson et al., 2016; Martínez-Alvarado et al., 2016; Baumgart et al., 2018; Grams et al., 2018;
Rodwell et al., 2018; Berman and Torn, 2019; Maddison et al., 2019; Sánchez et al., 2020). Accordingly, a misrepresentation
of WCBs in numerical weather prediction (NWP) and climate models may contribute to systematic forecast errors in the
large-scale flow so that information about the predictive quality of WCBs are desirable.

A systematic verification of WCBs in these models requires objective methods for the identification of WCBs which can be
automatically applied to large data sets. In early studies, WCBs were identified rather subjectively via cyclone-relative stream
lines on surfaces of constant wet-bulb potential temperature (e.g., Harrold, 1973; Browning and Roberts, 1994). In the absence
of diabatic processes, the cyclone-relative isentropic streamlines can also be used to represent trajectories. This assumption,
however, is barely justified in WCBs since their ascent is characterized on average by 20 K diabatic heating (Madonna et al.,
2014). To account for the cross-isentropic ascent, Wernli (1997) identified WCBs on the basis of kinematic forward trajectories
calculated from the three-dimensional wind field in gridded atmospheric data sets. They defined WCBs as coherent ensembles
of trajectories along which the specific humidity decreases in 48 hours by at least $12\,\mathrm{g\,kg^{-1}}$ or which ascend in 48 hours by
at least $620\,\mathrm{hPa}$. The trajectory-based definition is nowadays widely used and has significantly advanced the understanding of
WCBs and their effect on the large-scale flow (e.g., Eckhardt et al., 2004; Grams et al., 2011; Martínez-Alvarado et al., 2016). In
particular, the increasing spatio-temporal resolution of atmospheric reanalysis data sets as well as the increasing computational
power allowed evaluations of WCBs and their physical properties from a climatological perspective (e.g., Eckhardt et al., 2004;
Madonna et al., 2014).

At the same time, systematic evaluations concerning the predictive quality of WCBs in NWP or climate models based on
the trajectory approach are still rare which is likely due to the lack of the required input data in these data sets. Reliable
trajectory calculations require data at a high enough temporal ($\mathcal{O}(\sim 3\text{–}6\,\mathrm{hrs})$), horizontal ($\mathcal{O}(\sim 1°)$), and vertical ($\mathcal{O}(\sim 10\,\mathrm{hPa})$)
resolution (e.g., Stohl et al., 2001; Bowman et al., 2013) which are not provided in large NWP data sets (e.g., Hamill and
Kiladis, 2014; Vitart et al., 2017; Pegion et al., 2019) or climate projections (Eyring et al., 2016). In order to systematically
assess the representation of WCBs in data sets alike, Quinting and Grams (2021) introduced a statistical framework that
allows the identification of two-dimensional WCB footprints from Eulerian fields. Their statistical framework is based on
gridpoint-specific multivariate logistic regression models that predict the conditional probabilities of WCB inflow, ascent,
and outflow. Based on their approach it was recently shown that the European Centre for Medium-Range Weather Forecasts'
(ECMWF) sub-seasonal reforecast (Vitart et al., 2017) exhibit significant biases concerning the WCB occurrence frequency
and that reliable predictions of WCBs are not possible beyond 10 days forecast lead time (Wandel et al., 2021). Though the
logistic regression models reliably predict the occurrence of WCBs at instantaneous time steps, the regression approach comes
with certain limitations. First, a forward predictor selection revealed a spatial variability in terms of the optimal predictors at
different gridpoints. Accordingly, Quinting and Grams (2021) developed gridpoint-specific regression models. These models,
however, do not take into account the information of neighbouring grid points when predicting the occurrence of WCBs.
Second, the WCB stages of inflow, ascent and outflow are connected in a Lagrangian sense due to the time sequence in which





they occur. This temporal coherence is not considered by the regression models. Third, Quinting and Grams (2021) developed separate models for each meteorological season to account for the seasonal variability of WCB occurrence. Thus, modelled probabilities for dates close to the transition from one season to the next may exhibit discontinuities due to the changing regression coefficients.

To circumvent these three limitations, this study introduces an advanced statistical framework based on convolutional neural

network (CNN) models (e.g., Fukushima and Miyake, 1982) which are specifically designed to learn from data on spatial grids and to take into account the information from neighbouring grid points. By performing convolutional operations on an input map, CNNs find a compact representation of the spatial structures that produce the desired prediction. In meteorology, CNN models have recently been applied successfully to detect synoptic-scale structures such as fronts (e.g., Lagerquist et al., 2019), atmospheric rivers (e.g., Muszynski et al., 2019; Prabhat et al., 2021), extratropical cyclones (Lu et al., 2020; Kumler-

Bonfanti et al., 2020), dry intrusions(Silverman et al., 2021), and tropical cyclones (e.g., Matsuoka et al., 2018; Prabhat et al., 2021). In this research, the architecture of the CNN models is based on the UNet which is a semantic-segmentation model that was originally developed for medical images (Ronneberger et al., 2015). While this paper focuses on the development and evaluation of the CNN models, a companion paper shows the versatile applicability of the models to reanalyses and NWP data Quinting et al., 2021.

The paper is organised as follows: Section 2 introduces the predictor and predictand data on which the CNN models are built. The architecture of the UNet CNN models as well as the training process is described in Section 3. The performance of the models during Northern Hemisphere winter and summer is evaluated for the testing period in Section 4. To better understand which predictors provide most of the CNN models' skill, a permutation feature importance is conducted in Section 5. We end with concluding remarks and an outlook in Section 6.

## 2  Data


Binary labels (or "ground truth") of WCB inflow, ascent, and outflow are derived from the Lagrangian WCB trajectory data of Madonna et al. (2014) extended to 2016 by Sprenger et al. (2017). The data set is based on 48-h kinematic forward trajectories computed from ECMWF's Interim reanalyses (ERA-Interim; Dee et al., 2011) with the LAGRangian ANalysis TOol (LA-GRANTO; Wernli and Davies, 1997; Sprenger and Wernli, 2015). The ERA-Interim data are derived 6-hourly at 00, 06, 12,

18 UTC on all available model levels and are remapped from its original T255 spectral resolution to a regular $1° \times 1°$latitude–longitude grid. Possible WCB starting points are found by seeding trajectories globally from an equidistant grid every 80 km in the horizontal and vertically every 20 hPa from 1050 to 790 hPa at 00, 06, 12, 18 UTC. Only those trajectories are considered that, first, ascend in 48 h by at least 600 hPa and, second, are matched with an extratropical cyclone (Wernli and Schwierz, 2006) at least once during this 48-h period. We exclude tropical cyclones by not considering cyclones between 25°S and 25°N

for the matching with the potential WCB trajectories. Following Schäfler et al. (2014), all identified WCB parcel locations at a given time are binned into three vertical layers which are referred to as WCB inflow, WCB ascent, and WCB outflow. WCB inflow is defined as those WCB parcels being located below 800 hPa. The ascent stage, which typically occurs with a time-





lag of several hours after the inflow, considers those WCB air parcels between 800 and 400 hPa. All WCB air parcels above
400 hPa define the WCB outflow stage which occurs with a time-lag after the ascent stage. In a final step, the parcel locations

are gridded for each layer on a regular $1° \times 1°$latitude–longitude grid. Labeling grid points without/with WCB trajectory as 0/1
yields dichotomous dependent two-dimensional predictands for WCB inflow, ascent, and outflow, respectively.

Predictors are computed from nearly the same ERA-Interim data as used for the trajectory computation. The only difference
is that the computation of predictors is based on data at the 1000, 925, 850, 700, 500, 300, and 200 hPa isobaric surfaces and
not on all available model levels. This is due to the intention that the CNN models shall be applicable to climate projections

or reforecast data, for example of the sub-seasonal to seasonal prediction project data base (Vitart et al., 2017), which are only
available on this limited number of vertical levels. The four most important predictors for WCB inflow, ascent, and outflow were
identified in a stepwise forward selection approach by Quinting and Grams (2021) and are listed in Table 1. As an additional
fifth predictor, we include the 30-day running mean climatological occurrence frequency of WCB inflow, ascent, and outflow
centered on each calendar day which is based on 6-hourly data from the gridded Lagrangian WCB data set for the period 1

January 1980 to 31 December 2016. The purpose of using this fifth predictor is to account for the seasonal variation in WCB
occurrence frequency so that the same CNN models can be applied year-round. This avoids the need to develop one model per
season (Quinting and Grams, 2021). For each of the three WCB stages of inflow, ascent, and outflow a separate CNN model
is developed for the Northern Hemisphere with the predictors listed in Table 1 serving as input maps. These CNN models are
referred to as *standard models*.

## 110    3    UNet convolutional neural network

In this study, we use variants of the UNet CNN architecture (Ronneberger et al., 2015) which was originally designed to
process biomedical images but has been successfully applied in meteorological applications (e.g., Lebedev et al., 2019; Ayzel
et al., 2020; Weyn et al., 2020). The UNet is an encoder-decoder neural network architecture and consists mainly of two
paths (Fig.1), the contracting path (encoder) which down-scales the input map from its original resolution using convolutional

layers and pooling, and the expanding path (decoder) which up-scales learned patterns back to the original resolution using up-
sampling and convolutional layers. In the following, we provide information on the format of the input maps, the contracting
path, and the expanding path.

### 3.1    Input map

In a first step, the data introduced in Section 2 are split into training, validation, and testing data sets. An essential requirement

is that the training, validation, and testing data sets are statistically independent. A random sampling from the entire time period
to create the three subsets would likely lead to highly correlated data sets. For example, a sample from 00 UTC on one day could
fall into the training set and a sample from 12 UTC on the same day into the testing set. The 12 h time interval between the two
samples would be considerably shorter than the synoptic timescale on which WCBs evolve. To avoid statistical dependence,
we split the data into the three subsets as shown in Table 2. The training data, which comprise the period 01 January 1980





to 31 December 1999, are used to train the CNN models. Validation data are a comparably small subset of 5 years that allow to compare models with different settings on unseen data and to identify the best performing model. The testing data, which comprise the period 01 January 2005 to 31 December 2016, are used to evaluate the best performing models on unseen data (Section 3). Though predictors and predictands are available at 00, 06, 12, 18 UTC, we train and validate the CNN models with 12-hourly data (00, 12 UTC) for computational reasons. The computationally less expensive testing of the models is performed

on 6-hourly data (00, 06, 12, 18 UTC).

Each training sample consists of $M \times N \times P$ input maps and an $M \times N \times 1$ output map. The variable $M$ is the number of rows (latitudes), $N$ is the number of columns (longitudes), and $P$ is the number of channels (number of predictor variables; $P = 5$). The CNN models of this study contain at least four so-called max-pooling layers (see Section 3.2), each downsampling a map two times. Therefore, $M$ and $N$ have to be a multiple of $2^{n+1}$ (Ayzel et al., 2020), where $n$ is the number of max-pooling

layers. With $1° \times 1°$ horizontal grid spacing $M$ would be 91 for the entire Northern Hemisphere and thus not a multiple of $2^{n+1}$. Accordingly, we decided to select data from 6°S to 89°N ($M$=96) in the latitudinal direction. The North Pole at 90°N is excluded due to infinite gradients when computing some of the predictors in Table 1 via finite differences. To account for the circular nature of the data in the longitudinal direction at the international date line, input padding is performed (Shi et al., 2015; Schubert et al., 2019): we pad 44 grid points east and west of the date line which increases $N$ from 360 to 448. As a

result, the computing time needed for the model training increases. Still, it improves the results since without input padding the modelled probabilities would exhibit discontinuities along the dateline.

Prior to input padding each of the five predictor variables is normalized for each training sample to

$$x'_{i,j} = \frac{x_{i,j} - \overline{x}}{\sigma} \tag{1}$$

where $x_{i,j}$ is the original value, $\overline{x}$ denotes the area-weighted mean and $\sigma$ is the area-weighted standard deviation. The reasoning

behind the normalization is to prevent predictors with large values to cause large weight updates in the CNN during training (Section 3.4).

### 3.2 Contracting path

The default setting in this work is a contracting path with four blocks, each of which contains two convolutional layers (blue triangles in Fig.1). These layers transform the input maps into so-called feature maps using convolutional filters. The convo-

lutional filters are three-dimensional tensors of learnable weights with a certain spatial kernel size (kernel size = $3 \times 3$ in this study) and the third dimension equal to that of the input map. The filters convolve through the input maps grid point by grid point with stride = 1 in this study and perform a convolution defined as (e.g., Lagerquist et al., 2019)

$$\mathbf{X}_i^{(k)} = f\left[\sum_{j=1}^{J} \mathbf{W}_i^{(j,k)} * \mathbf{X}_{i-1}^{(j)} + b_i^{(k)}\right] \tag{2}$$

where $\mathbf{X}_i^{(k)}$ is the $k$th feature map in the $i$th layer, $\mathbf{X}_{i-1}^{(j)}$ denotes the $j$th feature map in the $(i-1)$th layer, $\mathbf{W}_i^{(j,k)}$ is the

convolutional filter, $J$ is the number of feature maps in the $(i-1)$th layer, $b_i^{(k)}$ is the bias for the $k$th feature map in the $i$th





layer, and $f$ is the activation function. We use the rectified linear unit (ReLU; Nair and Hinton, 2010) as activation function in order to add non-linearities to the convolutional layer output. This non-linearity is required since otherwise the CNNs would only learn linear relationships. The third layer of each block is a $2 \times 2$ max-pooling layer (orange triangles in Fig.1) which slides over each feature map with stride = 1 and takes the maximum of four numbers in the filter region of $2 \times 2$ grid points.

Accordingly, the feature maps are downsampled by a factor of 2. For example, the original size of the input map is $96 \times 448$ and after the first block, which contains one max-pooling layer, it is reduced to $48 \times 224$. The process of convolution and max pooling is repeated for each block. With each block, the number of filters doubles so that the models are able to detect the meaningful features of the input maps effectively. Each max-pooling layer is followed by a dropout regularization layer which aims to prevent overfitting (Srivastava et al., 2014). During dropout regularization, input units are randomly set to 0 with a

pre-defined dropout fraction at each step during training time. Though the effectiveness of dropout regularization in CNNs is still debated (e.g., Hinton et al., 2012; Ioffe and Szegedy, 2015), we decided to test the sensitivity of the results to the dropout fraction by varying it in the range from 0.0 to 0.3 at intervals of 0.05 (see Section 3.5). Dropout regularization is not used during validation and testing. Further, we apply batch normalization (Ioffe and Szegedy, 2015)[1] after each dropout layer which effectively reduces overfitting in CNNs and reduces the number of training steps.

## 3.3 Expanding path

In line with the contracting path the expanding path consists of four blocks, each of which contains three layers. The first layer is a transposed convolutional layer and serves the purpose to up-sample the feature maps from low to higher resolution. The kernel sizes are set to $3 \times 3$, and the stride is 2. The upsampling is followed by the second layer which first concatenates the feature maps from the contracting path to the expanding path (so-called skip connections), second applies a dropout function,

and third includes a convolutional layer with a kernel size of $3 \times 3$ and stride = 1. By including skip connections (black dashed arrows in Fig.1), high resolution information from the contracting path can be used in order to reconstruct high resolution feature maps in the expanding path. The third layer is a further convolutional layer with the same kernel size and stride as in the previous layer. After each expanding block the number of filters halves in contrast to the contracting blocks. The spatial dimensions double with each expanding block so that the size of the feature map is $96 \times 448$ after four expansions which is

the same size as the original input map.

The final output is generated with a convolutional layer (kernel size = $1 \times 1$; stride = 1; black triangle in Fig.1) which reduces the number of feature maps from 16 to 1. In contrast to all previous convolutional layers, the activation function is a sigmoid function yielding values between 0 and 1 so that the output can be interpreted as a conditional probability.

---

[1]During model training (Section 3.4), the training data set is divided into so-called batches. In brief, the batch size defines the number of training samples considered before updating the filter weights and batch normalization describes the process of normalizing the inputs maps in one batch prior to proceeding with the training.





## 3.4 Model training

Random initialization of the convolutional filters ensures that they all have different weights $\mathbf{W}_i^{(j,k)}$ and biases $b_i^{(k)}$. Accordingly, the different filters detect different features on the input map. The weights and biases of all convolutional filters are updated during iterative training via the Adam optimization algorithm (Kingma and Ba, 2015). The purpose is to minimize a loss function for classification which is in this study the binary crossentropy loss. It is defined as

$$L = -\frac{1}{N}\sum_{i=1}^{N} y_i \cdot log\hat{y}_i + (1 - y_i) \cdot log(1 - \hat{y}_i) \tag{3}$$

and is commonly used for binary classification tasks. $N$ is the number of scalar values in the model output, $\hat{y}_i$ denotes the probability that the $i$th example is a WCB, and $y_i$ is the corresponding target value (WCB yes or no).

The weights and biases are optimized using at most 20 training iterations (called epochs in the context of machine learning) with batch sizes ranging from 8 to 64. The initial learning rate of the Adam optimizer is set to $1 \times 10^{-3}$ and is reduced by a factor of 0.1 when the binary crossentropy does not improve over the course of 5 consecutive iterations. Further, the training

stops early if the binary crossentropy does not improve in 10 consecutive iterations.

## 3.5 Model setting optimization

In this section, we evaluate the performance of different models setups for the validation period (1 January 2000 to 31 December 2004) in order to find the optimal setting of parameters. A particular focus is on the hyperparameters *dropout fraction* and *batch size*. Further, we evaluate the effect of omitting the WCB climatology as predictor (4block_16filters in Fig. 2), adding

an additional fifth block to the UNet CNN (5block_WCBCLIM_16filters in Fig. 2), and increasing the number of initial filters from 16 to 32 (4block_WCBCLIM_32filters in Fig. 2). We try all 102 possible combinations of the parameters listed in Table 3.

In order to find the optimal configuration, we assess the model performance on the entire set of validation data in terms of the Matthews Correlation Coefficient (Matthews, 1975, MCC). The MCC is a balanced skill metric for binary verification tasks, even if the two classes are imbalanced as is the case with WCBs which occur at some grid points only in 1% of the cases. The

MCC is defined as

$$MCC = \frac{TP \times TN - FP \times FN}{\sqrt{(TP+FP)(TP+FN)(TN+FP)(TN+FN)}} \tag{4}$$

with TP, TN, FP, and FN being the number of true positives, true negatives, false positives, and false negatives. The MCC values range from –1 (total disagreement between prediction and observations) to 1 (perfect prediction). An MCC value of 0 indicates a random prediction. As the calculation of the MCC requires deterministic predictions (WCB yes or no), we evaluate it at different decision thresholds ranging from 0.05 to 0.95 at intervals of 0.05 above which the modelled probabilities are

set to one. The MCC is calculated gridpoint-wise but only at those grid points where the 30-d running mean climatological WCB occurrence frequency based on the period 1 January 1980 to 31 December 2016 reaches at least 1%. It should be noted that MCC values obtained from this gridpoint-wise evaluation are rather on the conservative side since slight offsets





between predictions and observations are punished unduly. Accordingly, object-based or neighbourhood-based evaluations (e.g. Lagerquist et al., 2019; Silverman et al., 2021) would yield even higher MCC values.

To begin with, we only focus on the *standard models* and their median MCC values (black, gray, blue and red lines Fig. 2) as a function of the decision thresholds. For all three WCB stages, WCB inflow (Fig. 2a), ascent (Fig. 2b), and outflow (Fig. 2c), the median MCC values for the different number of filters and blocks exhibit sensitivities of less than 5%. Even when not using the running mean WCB climatology as a fifth predictor, the median MCC does not decrease markedly (black line in Fig. 2). Larger sensitivities are found concerning the dropout fraction and batch size, especially for decision thresholds larger

than 0.5 as indicated by the wide range between the minimum (down-pointing triangles) and maximum (up-pointing triangles) MCC values. However, for the range of decision thresholds between 0.2 and 0.4 that yield the overall highest MCC, the range between the minimum and maximum MCC is less than 5% of the corresponding median MCC. For all three WCB stages, a *standard model* consisting of 4 layers and 32 filters yields the highest MCC (blue up-pointing triangles in Fig. 2). Of all *standard models*, the highest MCC values are reached for WCB ascent.

This result inspired us to test an additional CNN model configuration for the WCB stages of inflow and outflow: As outlined in the introduction, the inflow of a WCB precedes the ascent stage and the outflow lags the ascent stage. Accordingly, we decided to account for this relationship by replacing the fifth predictor of the *standard models* for inflow and outflow (30-d running mean WCB climatology) with the conditional WCB ascent probability predicted by the optimal WCB ascent model at a certain time-lag. Here we decided for a time-lag of 24 hours because the model is to be applied to forecasts of the sub-

seasonal to seasonal prediction project data base (Vitart et al., 2017), which are available 24 hourly. Thus, the fifth predictor is the conditional probability of WCB ascent 24 hours later (earlier) than the corresponding WCB inflow (outflow) time. These models are referred to as *time-lag models*. Indeed, the median MCC values for WCB inflow and outflow improve by almost 20% when taking the conditional probability of WCB ascent as a fifth predictor (4block_MIDTROP_32filters/green line in Figs. 2a, c). The highest median MCC for WCB inflow even exceeds that for WCB ascent. As for the *standard models* the

variations related to the parameter settings of dropout fraction and batch size are comparably small. The overall highest MCC values are reached with dropout fractions of 0.3 and batch sizes of 8 for inflow/ascent and 16 for outflow. It should be noted that the differences in terms of the MCC between the best and second best model are marginal. With a slightly different training period, slightly different dropout fractions and batch sizes may be considered optimal. Accordingly, the parameter testing is by no means comprehensive and additional parameter testing may yield better results. For the remainder of this study, we use

the *time-lag models* with 4 blocks, 32 filters, a dropout fraction of 0.3 and batch sizes of 8 for inflow/ascent and 16 for outflow (see headings in Figures 2a-c).

## 4   Model evaluation

The models are evaluated in terms of their reliability, biases, and skill for the entire testing period (1 January 2005 to 31 December 2016) though results are only shown for DJF and JJA, i.e., the seasons with highest and lowest climatological WCB





occurrence frequency, respectively (Madonna et al., 2014). Further, we compare the reliability and skill of the CNN models and the logistic regression models of Quinting and Grams (2021).

## 4.1 Reliability

The average agreement between the observed WCB frequencies and the modelled WCB probabilities for DJF and JJA is shown as reliability diagrams in Fig. 3. For this purpose, the predicted probabilities are divided into 19 regular bins from 0.05 to 0.95

and plotted against the observed frequencies in these bins. The reliability curve of a perfect model would follow the solid diagonal line in Fig. 3. A model overestimates (underestimates) the observed WCB frequency when the model's curve lies below (above) the solid diagonal line.

For DJF and JJA, the CNN models tend to slightly overestimate the observed WCB inflow frequency at modelled probabilities greater than 0.3 and 0.1, respectively (Fig. 3a). The overestimation is more pronounced in JJA. Still, the reliability

curve lies inside the ±10% range of a perfect model. The CNN models clearly outperform the logistic regression models for modelled probabilities greater than 0.3 in JJA and greater than 0.5 in DJF. As for the logistic regression models in Quinting and Grams (2021), the CNN models perform best for WCB ascent (Fig. 3b). For DJF and JJA, the reliability curve nearly matches the solid diagonal line of the perfect model. For WCB outflow, the reliability curves lie within the ±10% range of the perfect model during DJF and JJA but slightly above the perfect model (Fig. 3c). This indicates that the CNN model underestimates

the observed frequencies. Still, the CNN model is more reliable than the logistic regression models which overestimate the observed frequencies considerably for modelled probabilities greater 0.5 and 0.7 during JJA and DJF, respectively.

## 4.2 Model bias

The evaluation of the bias and skill (see Section 4.3) of the CNN models requires categorical/deterministic predictions (WCB yes or no). Therefore, the probabilistic CNN model predictions need to be categorized by applying a decision threshold above

which a modelled probability is considered as WCB inflow, WCB ascent, or WCB outflow. Following (Quinting and Grams, 2021), the decision threshold is chosen to be gridpoint-dependent and to minimize the climatological bias of the models at each grid point. Here, bias is defined as the difference between the trajectory-based climatological WCB frequency and the CNN-based climatological WCB frequency in the testing period (1 January 2005 – 31 December 2016). The decision thresholds are assessed as follows:

1. For each day of the year we compute a 90-day running mean Lagrangian WCB climatology. Although the assessment of reliability, bias, and skill is performed for the testing period (1 January 2005 – 31 December 2016), the climatological WCB frequency used to define the decision thresholds is based on the entire period (1 January 1980–31 December 2016) in order to account for possible long-term variations of the WCB occurrence frequency.

   2. We then loop over a decision threshold $0 < p_{WCB} < 1$ at intervals of $0.01$ above which the conditional probability

275       predicted by the CNN is set to 1.





3. For each day of the year and for each value of the decision threshold we compute a 90-day running mean climatology based on the CNN model predictions.

4. For each day of the year and each grid point, we determine the optimal $p_{WCB}$ which is the one that produces the lowest bias between the trajectory-based and CNN-based WCB climatology.

The purpose of calculating a decision threshold $p_{WCB}$ for each day of the year is to account for seasonal variations of the modelled probabilities. For WCB inflow, ascent, and outflow the decision threshold is highest in winter and lowest in summer (not shown).

In the following, we analyse to what degree the climatological occurrence frequency of WCB inflow, ascent, and outflow based on gridded trajectories is represented by the CNN-based approach (Fig. 4) in the testing period. By design of the decision
threshold above which a predicted probability is considered as WCB, the observed frequency and that of the regression model coincide well. During DJF, the model bias for WCB inflow, ascent, and outflow is less than 2% at all grid points. Also in JJA, the biases are of similar magnitude except for the region related to the Asian monsoon over Northern India (Figs. 4b,d,f). Here, the frequency bias reaches up to 3%.

### 4.3  Model skill

The skill of the CNN models is quantified in terms of the MCC during the testing period. For WCB inflow and during DJF, the models' skill is highest in the climatologically most frequent inflow regions over the western to central North Pacific and the western to central North Atlantic (Fig. 5a). Especially in regions where the climatological WCB inflow frequency exceeds 10%, the MCC reaches values of 0.6 to 0.7. Compared to the logistic regression model the MCC improves in most regions by at least 50% (Fig. 5b) which corresponds to an absolute improvement of the MCC by 0.3. The improvements are most
pronounced in regions of comparably low climatological WCB inflow frequency, indicating that the CNN models also perform well on data with a limited availability of training samples.

The MCC values for WCB ascent are of similar magnitude (Fig. 5c). Again, the highest MCC values greater than 0.6 are collocated with the climatological occurrence frequency maxima over the Atlantic and western to central North Pacific. Even in regions where the climatological occurrence frequency is on the order of 1%, the MCC exhibits values of at least 0.5. The
improvement of the MCC compared to the logistic regression model is positive everywhere (Fig. 5d), but smaller than for WCB inflow and outflow. Still, the mean MCC for WCB ascent is the second highest of the three WCB stages (cf. Fig. 2).

The MCC for WCB outflow is lower than for WCB inflow and WCB ascent. Highest MCC values of 0.6 to 0.7 are found over eastern North America, the Labrador Sea, and over the western to central North Pacific (Fig. 5e). Thus, as for WCB inflow and ascent, the regions of highest model skill are collocated with regions exhibiting the highest climatological WCB outflow
frequency. The relative improvement compared to the logistic regression models is particularly pronounced in the regions with the climatologically lowest WCB outflow occurrence frequency. The absolute value of the MCC exceeds that of the regression models by more than 0.25 which corresponds in some locations to a relative increase of more than 125% (Fig. 5f).





During boreal summer (JJA) the frequency of WCB inflow, ascent, and outflow is considerably lower than during DJF. WCB inflow occurs most frequently over central North America, the western North Atlantic, and over East Asia to the central North

Pacific (Fig. 6a). A further maximum north of India is related to the Asian monsoon. On average the MCC for WCB inflow tends to be lower during JJA than during DJF. Still, in regions with a climatological frequency of more than 2%, the MCC reaches values of 0.4 to 0.6 which corresponds to a relative improvement of 50% over the western North Pacific and more than 125% over central North America compared to the logistic regression models.

The MCC values for WCB inflow and WCB ascent are of similar magnitude. Although the climatological frequency is only

about 2% in the storm track regions of the North Atlantic and the North Pacific, the MCC still exceeds 0.5 in these areas (Fig. 6c). The absolute value of MCC improves by more than 0.15 compared to the regression models in most regions which corresponds to a relative increase of 25–50% (Fig. 6d).

Also in JJA, the MCC is lower for WCB outflow than during DJF. In regions where the WCB outflow occurrence frequency exceeds 2%, the MCC mostly exceeds values of 0.4 (Fig. 6e). However, in regions of climatologically low WCB outflow

occurrence frequency the MCC is on the order of 0.2 to 0.3 which is most likely related to the comparably small training sample size in those areas. Especially in high latitudes over the North Pacific and over the entire North Atlantic, the MCC improves by more than 100% compared to the logistic regression models at most grid points.

## 5  Identifying the most relevant dynamical footprint of WCBs by interpreting feature importance in the CNN models

Prior to the logistic regression model development, Quinting and Grams (2021) identified the best predictors for the three

stages of WCB inflow, ascent, and outflow via stepwise forward selection. Here, we take the opposite approach and use the fully developed CNN models to address the question which of the predictors used for the *time-lag models* have the largest impact on the predictions and are thus most characteristic for each of the WCB stages. Following Breiman (2001) and Gagne et al. (2019), we choose a model-agnostic interpretation method known as permutation feature importance which treats the model as a black box and only operates on the inputs and outputs. Permutation feature importance ranks predictors based on

how randomizing their values affects the models' skill. More specifically, we take the models' MCC skill of Section 4.3 as reference and compare it to the MCC skill of predictions where predictor P1 (Table 1) for each WCB stage is sampled at a random date from the testing period. The remaining four predictors are sampled at the exact same date. This process is repeated for predictors 2 to 5 so that the skill of 5 different predictions in terms of the MCC can be compared. The larger the decrease in MCC, the higher the importance of the corresponding predictor. Though the normalization of the input data should reduce

the effect of seasonal variations, we still take the random dates from a window of 30 days around the actual date.

According to the permutation feature importance, the most important predictor variables for WCB inflow during DJF is the conditional probability of WCB ascent 24 hours later (referred to as MIDTROP in Figs. 7a,b). The average skill decrease in terms of the MCC is twice as high as the decrease when perturbing the 1000-hPa moisture flux convergence and 850-hPa meridional moisture flux (Fig. 7a). It is only at the edges of the climatologically most active WCB inflow regions where

moisture flux convergence and the meridional moisture flux are identified as most important predictors. Also during JJA, the





conditional probability of WCB ascent with a time-lag of 24 hours is the most important predictor for WCB inflow (Fig. 8a,b). It is followed by the 1000-hPa moisture flux convergence and the 850-hPa meridional moisture flux. The 700-hPa thickness advection as well as the 500-hPa moist PV are of minor importance in both seasons. That the conditional probability of WCB ascent with a time lag of 24 hours is the most important predictor for WCB inflow is in line with the original trajectory-based

definition where a temporal relation between the two stages is given by definition. The comparably high importance of variables related to moisture flux is in line with the findings for the logistic regression models but also with the general concept of WCB inflow which is typically characterized by strong moisture flux convergence and bands of high water vapor transport (Wernli and Davies, 1997; Dacre et al., 2019).

    For WCB ascent, a permutation of the 850-hPa relative vorticity leads to the strongest decrease in model skill (Figs. 7c,

8c). In particular over the western North Pacific and the western North Atlantic the MCC decreases to values near 0 when perturbing the relative vorticity field (Figs. 7d, 8d). These findings are the same for DJF and JJA. During WCB ascent, relative vorticity is redistributed via stretching so that cyclonic vorticity increases in the lower troposphere (Binder et al., 2016). Thus, the overall importance of relative vorticity for WCB ascent is in line with physical considerations. The decrease of the MCC for WCB ascent due to permutations of the 300-hPa thickness advection, 850-hPa meridional moisture flux, and 700-hPa relative

humidity exhibits similar values. The seemingly least important predictor is the climatological WCB occurrence frequency with a median decrease in MCC close to zero. However, one should keep in mind that the random dates are taken only from a window of 30 days around the actual date. By doing so, the importance of the climatological WCB occurrence frequency for predicting the seasonal cycle of WCB activity is likely underestimated.

    The almost equally most important predictor variables for WCB outflow during DJF are the conditional probability of

WCB ascent 24 hours before and the 300-hPa relative vorticity (Figs. 7e,f). Interestingly, over the North Pacific the WCB ascent predictor is most important at nearly all grid points while over the North Atlantic the 300-hPa relative vorticity is the most important predictor at about half of all grid points. During the summer months, the 300-hPa relative vorticity becomes less important (Fig. 8e). At nearly all grid points the conditional probability of WCB ascent is the most important predictor (Fig. 8f). It is only in regions with a climatologically lowest WCB frequency where the 300-hPa relative vorticity and the

300-hPa irrotational wind speed are still the most important predictors. The importance of the conditional probability of WCB ascent with a time lag of –24 hours coincides with the trajectory-based WCB identification where this relation is given by definition. The importance of the 300-hPa relative vorticity is most likely related to the fact that WCB outflow is most often found in upper-tropospheric anticyclonic ridges (e.g., Pomroy and Thorpe, 2000; Grams et al., 2011).

## 6   Conclusions

In this study, we introduce a UNet CNN that aims to identify WCB footprints from Eulerian fields which are available from NWP and climate models. For each of the WCB stages of WCB inflow, ascent, and outflow a separate CNN model is developed. The CNN-based framework is trained for the Northern Hemisphere on 20 years of gridded trajectory-based WCB data derived from ERA-Interim using the same physical predictors as in Quinting and Grams (2021). The climatological occurrence





frequency of WCB inflow, ascent, and outflow serve as an additional predictor for the respective WCB stage. With these pre-

dictors, the UNet *standard models* consisting of 4 layers with an initial set of 32 filters yield the best results for WCB ascent. Sensitivities to the hyperparameters dropout fraction and batch size are found to be small. Given that the CNN model performs best for the WCB ascent stage, we make use of the temporal succession of the three WCB stages to predict WCB inflow and outflow. For WCB inflow and outflow, the fifth predictor in the *standard models* is replaced by the conditional probability of WCB ascent predicted with CNN model at a time-lag of 24 h and –24 h, respectively. With this approach, the improvement of

the CNN models for inflow and outflow is considerably larger than any variations of the hyperparameters so that we consider these models as optimal. The importance of the time-lagged conditional probability of WCB ascent as a predictor for WCB inflow and outflow is confirmed by the model-agnostic permutation feature importance. Further important predictors related to moisture flux for WCB inflow or relative vorticity for WCB ascent and outflow are in line with previous trajectory-based studies and highlight the capability of the CNNs to identify WCBs based on dynamical features that are in agreement with the

general concept of WCBs.

The CNN models for WCB inflow, ascent, and outflow are evaluated for an unseen testing period covering 1 January 2005 to 31 December 2016. For all three WCB stages, the models' reliability is within the 10% interval around the reliability of a perfect model. The models reach a similar reliability during boreal summer and winter. Most notably, the models outperform the logistic regression models of Quinting and Grams, 2021 which tend to overestimate the frequency of WCBs at any of the

three stages. The modelled probabilities are converted to dichotomous predictions by determining a decision threshold such that the climatological bias of the models is minimized. For all three stages, the models reach the highest skill in terms of the Matthews correlation coefficient in the midlatitude storm tracks regions, i.e., in regions where the climatological occurrence frequency of WCBs is highest. Compared to the logistic regression models, the relative skill improvement reaches up to 100%.

Our study demonstrates that deep learning allows transferring a sophisticated diagnostic, which relies on high resolution

data and considerable computing time, into a reliable and almost unbiased tool, which works on coarser data with significantly less computing time. This opens promising pathways how to use machine learning for process-oriented studies on big data sets such as ensemble NWP reforecasts or climate model projections that were so far inaccessible due to diagnostic constraints. For example, the CNN-based WCB models can be used to investigate the representation of the climatological frequency of WCBs in these data sets but also of the link of WCB activity and midlatitude synoptic systems such as cyclones or blocking.

The feature importance opens ways to pin down biases in the WCB frequency to biases in the predictor variables. Moreover, the high skill in instantaneous WCB identification allows to use the WCB diagnostic as an additional inexpensive feature identification tool in case studies. Examples of such applications are discussed in Part II of this study. It includes an analysis of the climatological link between WCBs and cyclones and blocking, an analysis of WCB frequency biases in ECMWF's operational ensemble forecasts, as well as examples showing the versatile applicability of the CNN-based models to modelling

systems other than ECMWF.

Ultimately, we do not argue that deep learning approaches should and can replace sophisticated diagnostic tools. Rather they should be used in concert with the latter by first establishing a fundamental understanding of physical processes based on an in-depth investigation of data sets with high spatio-temporal resolution. In a second step, a companion deep learning diagnostic



such as presented in this study facilitates testing the representation of such processes in larger data sets from NWP and climate
models.

*Code and data availability.* The exact version of the *time-lag models*, the decision thresholds, the 30-d running-mean trajectory-based WCB climatology as well as code to process the input data for the models are provided via the following repository https://git.scc.kit.edu/nk2448/wcbmetric_v2.git and archived on Zenodo (https://doi.org/10.5281/zenodo.5154980). ERA-Interim data are freely available at https://apps.ecmwf.int/datasets/data/interim-full-daily. LAGRANTO source code and documentation can be downloaded from http://www.lagranto.ethz.
ch.

*Author contributions.* JQ developed the models and conducted the model evaluation. CMG provided the initial idea. JQ and CMG jointly discussed and interpreted the results and prepared the manuscript.

*Competing interests.* The authors declare that they have no conflict of interest.

*Acknowledgements.* This work was funded by the Helmholtz Association as part of the Young Investigator Group "Sub-seasonal Predictabil-
ity: Understanding the Role of Diabatic Outflow" (SPREADOUT, grant VH-NG-1243). The research was partially embedded in the subprojects A8 and B8 of the Transregional Collaborative Research Center SFB/TRR 165 'Waves to Weather' (https://www.wavestoweather.de) funded by the German Research Foundation (DFG). Sincerest thanks to the Atmospheric Dynamics group at ETH Zurich in particular to Michael Sprenger and Heini Wernli for sharing the trajectory-based WCB data. We are grateful to Sebastian Lerch at KIT for an inspiring workshop that motivated the implementation of the CNN and to the Large-scale dynamics and predictability group at KIT for helpful
discussions. ECMWF, Deutscher Wetterdient, and MeteoSwiss are acknowledged for granting access to the ERA-Interim data set.



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



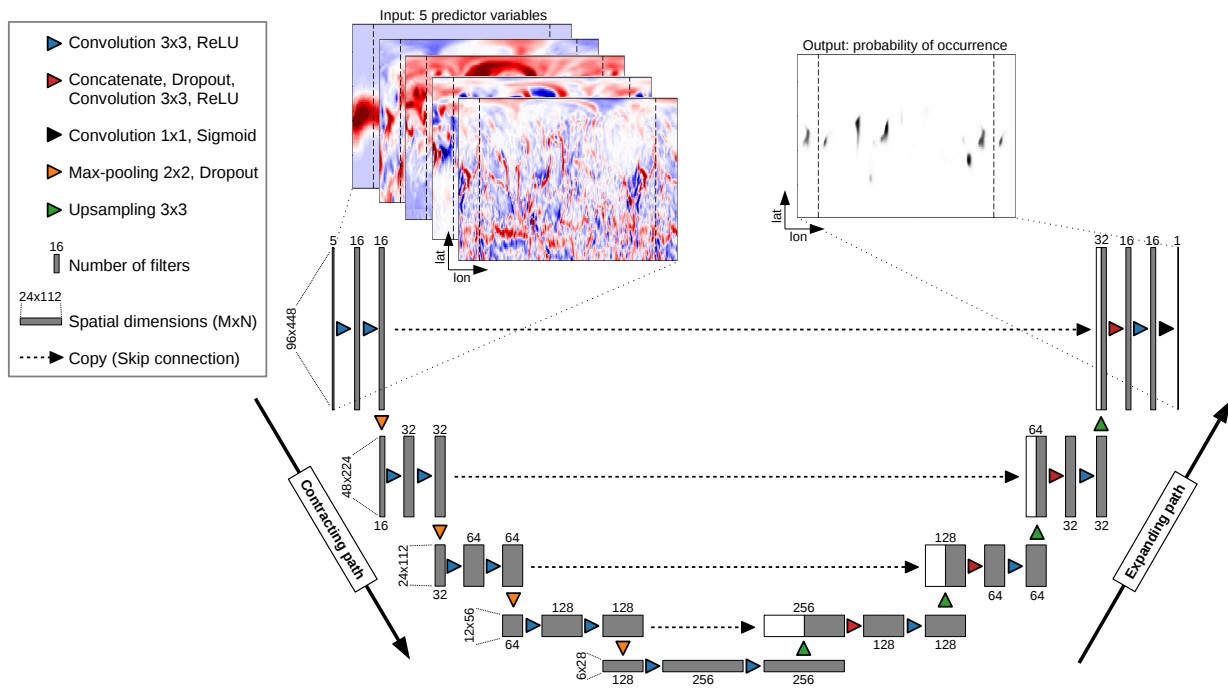

**Figure 1.** The architecture of the UNet which follows an encoder-decoder structure with a contracting path and an expanding path. Both paths consist of four blocks each with three layers. A final $1 \times 1$ convolutional layer reduces the number of feature maps from 16 to 1. See main text for detailed explanations.



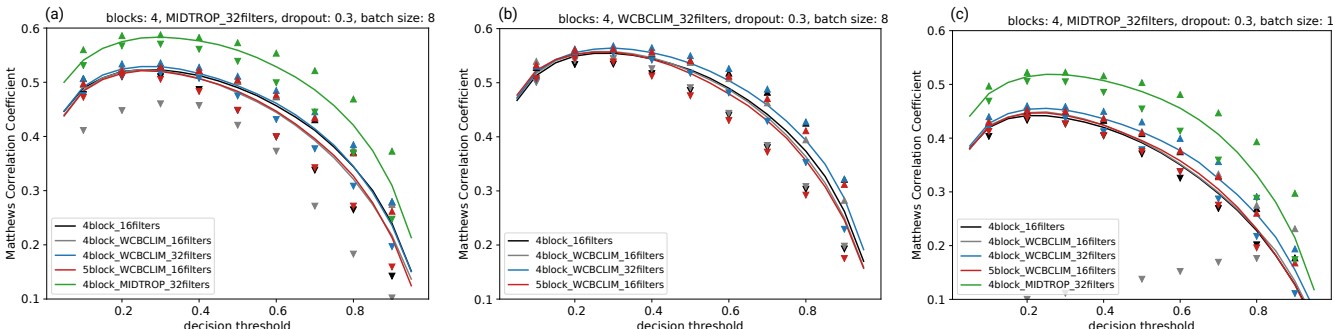

**Figure 2.** MCC values for (a) WCB inflow, (b) WCB ascent, and (c) WCB outflow as a function of the decision thresholds for different parameter settings described in Section 3.5 and listed in Table 3. MCC values are averaged over those grid points where the 30-d running mean climatological WCB frequency reaches at least 1%. Lines are median over all experiments with varied dropout fraction and batch size. Down- and up-pointing triangles denote their minimum and maximum values, respectively. The model configuration reaching the highest MCC is given in the title of each subplot.





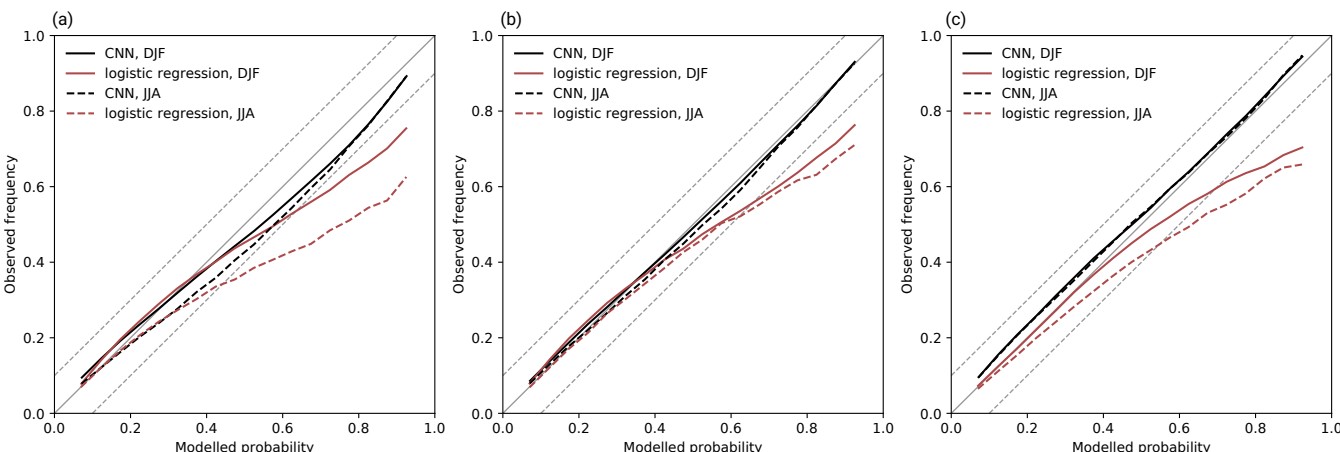

**Figure 3.** Reliability diagrams for (a) inflow, (b) ascent, and (c) outflow during DJF (solid lines) and JJA (dashed lines). The black curves represent the reliability of the CNN models and the red curves represent the reliability of the logistic regression models (Quinting and Grams, 2021). Modelled probabilities (x-axis) and observed frequencies (y-axis) are binned into 19 bins based on the modelled probabilities. The perfect modelled probability and a ±10% interval about the perfect model is shown by the solid and dashed gray diagonals, respectively.





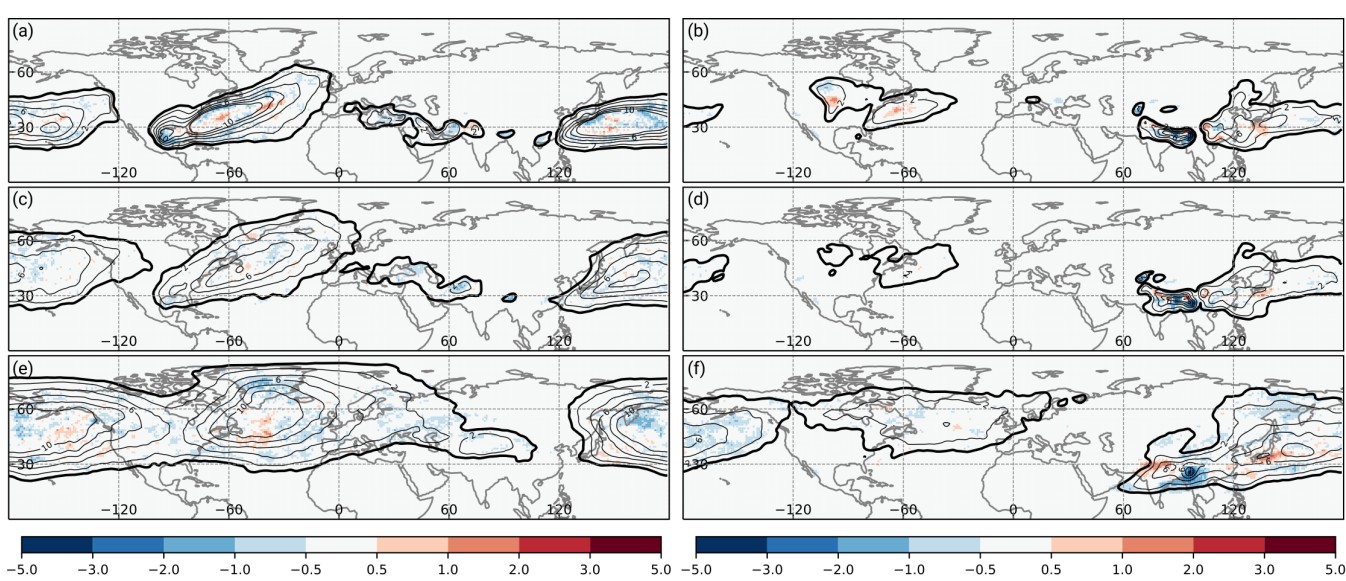

**Figure 4.** Climatological occurrence frequency bias for WCB (a, b) inflow, (c, d) ascent, and (e, f) outflow (shading is absolute frequency bias in percent) of the CNN models compared to the trajectory-based climatology (thick black contour at 1%; thin black contours every 2%). Panels (a, c, e) are for DJF in the period 1 December 2005 to 29 February 2016 and panels (b, d, f) are for JJA in the period 1 June 2005 to 31 August 2015.





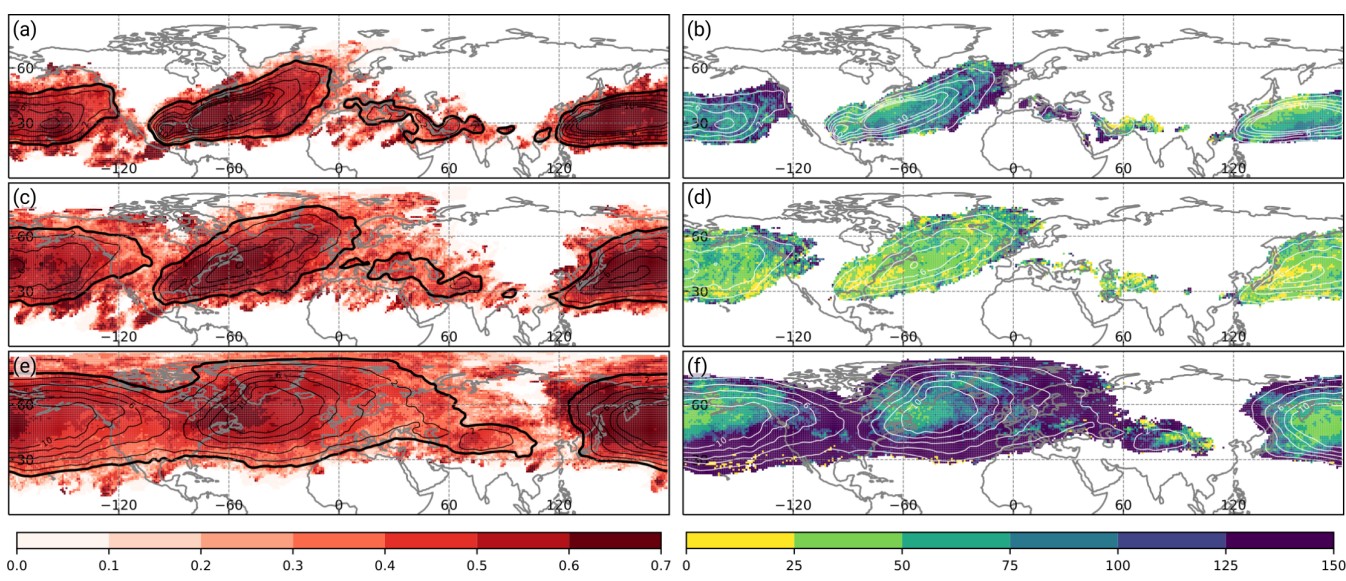

**Figure 5.** Matthews correlation coefficient of the CNN models during DJF for (a) WCB inflow, (c) WCB ascent, and (e) WCB outflow (shading). Relative difference in terms of the Matthews correlation coefficient between the CNN model and the logistic regression models in Quinting and Grams (2021) for (b) WCB inflow, (d) WCB ascent, and (f) WCB outflow (shading in %). Relative difference is only shown at grid points where the climatological Lagrangian WCB frequency reaches at least 1% since the logistic regression models are not available at grid points with a lower climatological frequency. Contours denote the climatological Lagrangian WCB frequency (thick black contour at 1%; thin black and white contours every 2%) for the respective WCB stage. All panels are shown for DJF in the period 1 December 2005 to 29 February 2016.





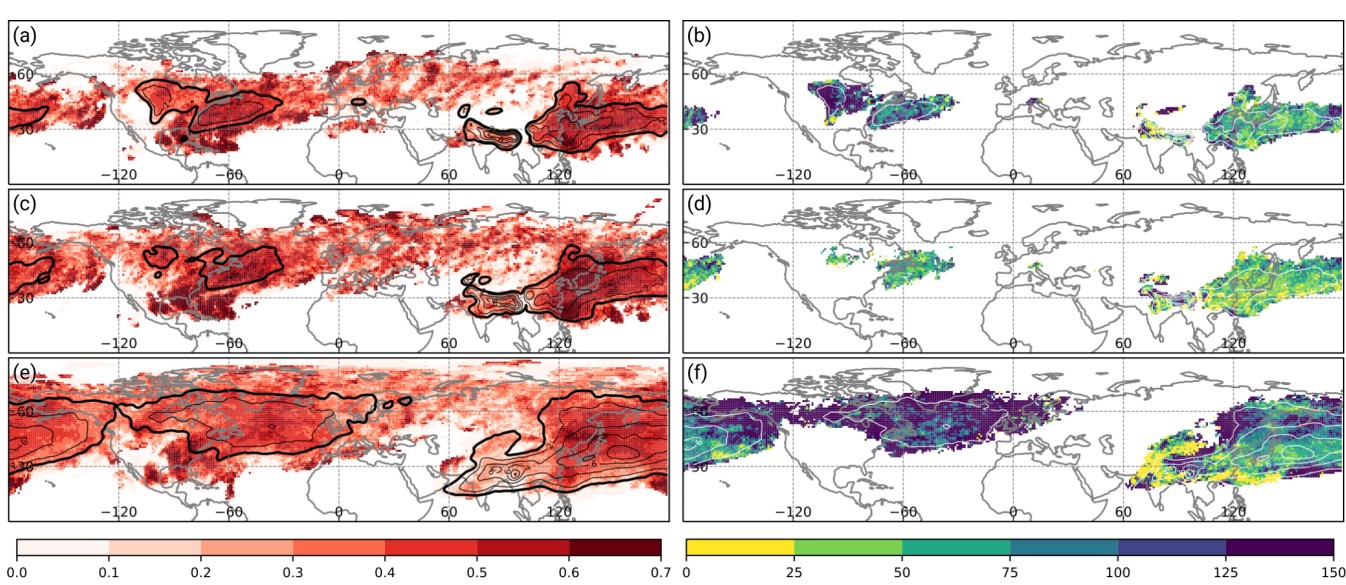

**Figure 6.** As in Fig. 5 except that all panels are shown for JJA in the period 1 June 2005 to 31 August 2015.





**Figure 7.** Feature importance scores in terms of the reduction in MCC for (a, b) WCB inflow, (c, d) WCB ascent, and (e, f) WCB outflow. Box and whisker plots in (a, c, e) show the median (vertical lign), interquartile range (boxes), and the 5 and 95 percentiles of the decrease in MCC over all grid points at which the climatological occurrence frequency reaches at least 1%. Shading in (b, d, f) shows the decrease in MCC when perturbing the most important predictor. The darker the shading (at intervals of 0.2) the greater the decrease of the MCC. Black contours indicate the MCC of the reference prediction at intervals of 0.1.





**Figure 8.** As in Fig. 7 except that all panels are shown for JJA in the period 1 June 2005 to 31 August 2015.



**Table 1.** The most important predictors for WCB inflow, ascent, and outflow as identified by Quinting and Grams (2021). These predictors are used in the CNN *standard models*. The abbreviations *P* and *rm* stand for *predictor* and *running mean*, respectively.

| P | WCB inflow | WCB ascent | WCB outflow |
|---|---|---|---|
| 1 | 700-hPa thickness advection (THA) | 850-hPa relative vorticity ($\zeta$) | 300-hPa relative humidity (RH) |
| 2 | 850-hPa meridional moisture flux (MFLY) | 700-hPa relative humidity (RH) | 300-hPa irrotational wind speed (wspd$_\chi$) |
| 3 | 1000-hPa moisture flux convergence (MFLCON) | 300-hPa thickness advection (THA) | 500-hPa static stability ($\sigma$) |
| 4 | 500-hPa moist potential vorticity (MPV) | 500-hPa meridional moisture flux MFLY | 300-hPa relative vorticity ($\zeta$) |
| 5 | 30-d rm inflow climatology (WCBCLIM) | 30-d rm ascent climatology | 30-d rm outflow climatology |



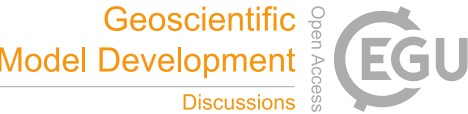

**Table 2.** Temporal data coverage of training, validation and testing data for the CNN models.

| Data set | Time Period | Number of samples |
|---|---|---|
| Training | 0000 UTC 1 Jan 1980 – 1200 UTC 31 Dec 1999 | 14610 (12 hourly data) |
| Validation | 0000 UTC 1 Jan 2000 – 1200 UTC 31 Dec 2004 | 3654 (12 hourly data) |
| Testing | 0000 UTC 1 Jan 2005 – 1800 UTC 31 Dec 2016 | 17532 (6 hourly data) |





**Table 3.** Parameters that are used to find the best parameter setup for the *standard models*.

| Parameter | Values |
| --- | --- |
| Number of filters/blocks | 16/4*, 16/4, 16/5, 32/4 |
| Batch size | 8, 16, 32, 64 |
| Dropout fraction | 0.0, 0.05, 0.1, 0.15, 0.2, 0.25, 0.3 |

* In these experiments the WCB climatology of WCB inflow, ascent, and outflow is omitted as predictor.