# Peer review of "EuLerian Identification of ascending Air Streams (ELIAS 2.0) in Numerical Weather Prediction and Climate Models. Part I: Development of deep learning model"

_Geoscientific Model Development, 2021_

## Author Response (AR1)

**Response to Reviewer 1**

The authors have developed a model for identifying warm conveyor belts in gridded datasets that do not have the required output for calculating trajectories. The model used is based on convolutional neural networks. The model is trained and tested using a limited set of variables from ERA Interim against trajectory calculations using a more complete dataset. The model shows good skill in identifying warm conveyor belts and significant improvement on the previous logistic regression models used by the authors.

The paper is well presented and fully justifies the conclusions. The results of this paper are very significant and it will be interesting to see the warm conveyor belt perspective in applications such as seasonal forecasting and climate modelling where the trajectory approach was not possible. I recommend the paper is accepted as is.

Dear Reviewer,

We are grateful for your very positive feedback on our manuscript. We very much hope that the novel diagnostic will be useful for the wider seasonal forecasting and climate modelling community and look forward to the first diagnostic applications.

Kind regards,
Julian Quinting & Christian Grams

**Response to Reviewer 2**

This study offers a well thought out alternative to a previous simple statistical model (logistic regression), and represents a significant improvement over the previous implementation. The Convolutional Neural Network (CNN) allows the consideration of non-local spatially dependent predictor/predictand variables, and is readily more interpretable than the logistic regression model. Generally, this is a strong paper, and subject to the minor comments below, the manuscript should be published. My main concern is with the with the generation of figure 4, which I detail in the minor comments below. Additionally, please add a section on limitations of the method, caveats, and future improvements that could be applied to this work.

Dear Reviewer,

We are very grateful for your overall positive feedback and the thought-provoking comments on our manuscript. We agree that the method still comes with certain limitations, and ideas are around for future improvements. Thus, we include a paragraph in the revised manuscript discussing these aspects (l. 405-411). In the following, we respond point by point to your minor comments. Our responses are highlighted in blue.

Kind regards,

Julian Quinting and Christian Grams

**Minor Points**

L55-L64 This is a good justification for a computer vision based machine learning approach.

Thank you for this positive feedback. We now include the key-word „computer vision based machine learning approach" in this part of the manuscript so that it connects directly to the next paragraph of the manuscript. The sentence now reads "It is primarily these two limitations that motivate the use of a computer vision based machine learning approach" (l. 64).

L66-67 this feels like a bit of a misrepresentation, I would change to say "CNNs identifying salient features in the input space which influence the desired prediction."

We fully agree. We changed the manuscript accordingly. The sentence now reads "By performing convolutional operations on an input map, CNNs identify salient features in the input space which influence the desired prediction" (l. 67).

L72. I would say that it is "originally designed" as a semantic-segmentation model, as it's applications are now much further reaching.

Thanks for this comment. We modified the manuscript as suggested. The sentence now reads "In this research, the architecture of the CNN models is based on the UNet which is originally designed as a semantic-segmentation model for medical images" (l. 71).

L106 How much model degradation occurs without this 5$^{th}$ predictor? Figure 2 seems to indicate that the seasonality is not a big factor, as much as including time-lagged ascent information. Figure 7. Confirms it is not a factor. This seems like something that needs to be explored or commented on further. Is this due to the normalization around the date of interest, and the selection of data around the forecast date. I think it is worth testing whether this variable affects the final skill of the model when you are not selecting data in a 30-day randomized window. Or de-emphasize this line in the introduction in general, as you immediately remove this variable as a predictor.

Thank you for this comment. We absolutely agree that Figs. 2 and 7 indicate that the 5th predictor is of minor importance for the models' skill. When designing the CNN-based models we hypothesized to see a benefit when incorporating the climatological occurrence frequency as a predictor. When evaluating the feature importance for a single season only as we do here, the 5$^{th}$ predictor indeed seems to be of minor importance. However, when we determine the feature importance across two seasons (e.g., summer and winter) the importance of the 5$^{th}$ predictor increases slightly. We now de-emphasize the aspect of the 5$^{th}$ predictor in the introduction and explain in Section 2: "Although the fifth predictor is of minor importance when considering a single season (see Section 5), its purpose is here to account for the variation in WCB occurrence frequency across different seasons so that the same CNN models can be applied year-round." (l. 106).

L157. The non-linearity is not necessarily required. Has it been tested to use linear activations? This would give you an idea of the linearity of the actually predictor/predictand relationship. You have two competing predictor improvements in this model (compared to local logistic regression) 1) the addition of a spatial component via convolution 2) the nonlinear predictor/predictand relationship. It would be good to test what is a bigger factor for model improvement, my inkling is the spatial information is more valuable.

Thank you very much for this interesting comment. Indeed, the logistic regression model suggests that a linear relationship exists between predictors and predictands. Since we have not tested yet using linear activations, we followed your suggestion and tested to what degree the models' skill changes with this approach. With linear activation functions the models' skill for WCB ascent in terms of the MCC decreases at most grid points (cf. Fig. 1 of this document and Fig. 5 of the original manuscript). Highest values exceeding 0.6 occur over the western North Atlantic and western North Pacific. The CNNs with linear activation functions still have higher skill than the logistic regression models though the improvement has decreased to 0-25% in most regions. As a reminder, the improvement with nonlinear activation functions was on the order of 25-50% in most regions. Thus, our take-home message from the additional analysis is that it is both the spatial component via convolution and the nonlinear activation function that lead to the improvement compared to the logistic regression models. Though we did decide to not include Fig. 1 of this document in the manuscript, we mention this result in the concluding discussion by stating "Compared to the logistic regression models, the relative skill improvement reaches up to 100% which is due to both the addition of spatial information via convolution and the non-linear activation functions which account for non-linear relationships between the predictors and predictands" (l. 402).

[Figure]

*Fig. 1. As Fig. 5 of the manuscript but shown here only the MCC for WCB ascent in (top row) DJF and (bottom row) JJA with linear activation functions in the CNN.*

L165. The debate over the efficacy of dropout is distracting to this paper. I would take it out.

We removed this part of the sentence in order to not distract from the main content of the paper.

L208. Please specify what dataset the MCC threshold tuning (0.05-0.95) tuning was done on.

The threshold tuning was done on the validation data. We now include this information in the manuscript (l. 209).

L245. Readers would benefit from a quick summary of Quinting and Gram's (2021) logistic regression model.

Thanks for this suggestion. We now provide a more detailed description of the logistic regression models in the introduction of the revised manuscript (l. 48ff).

L255 The authors do not define why +- 10% is considered perfectly reliable (nor do they test via any subsampling), either justify this more clearly, or I would suggest adopting the Bröcker and Smith reliability diagram framework (Bröcker, J., & Smith, L. A. (2007). Increasing the Reliability of Reliability Diagrams, *Weather and Forecasting*, *22*(3), 651-661.)

Thank you very much for pointing us to the improvements of the reliability diagram by Bröcker and Smith (2007). We adopted their approach by including consistency bars calculated via consistency resampling with 1000 iterations with replacement. Though the overall interpretation of the reliability diagrams remains nearly unchanged, the inclusion of the consistency bars indeed shows that the +-10% interval indeed was chosen too high to indicate perfect reliability. We corrected the discussion of the reliability diagram in Section 4.1 where necessary.

L270-280 Can you justify why this process should be performed on the testing dataset and not the validation dataset? It appears as if this is tuning a hyperparameter, and you are increasing your model bias skill on the testing data. It seems like the thresholds should be determined on the validation data as you don't plan on running the expensive Lagrangian framework model when implementing this CNN in the future. This seems concerning for this figure.

We agree that it is indeed more intuitive to determine the thresholds on the validation data. The issue we encounter here, however, is that due to the low climatological occurrence frequency of WCBs the thresholds become spatially highly variable. Accordingly, two neighboring grid points may be classified as non-WCB and WCB despite exhibiting nearly equal conditional probabilities. Thus, taking the longer testing data period yields spatially less variable thresholds. Further, with the relatively short  validation period the results may be affected by possible long-term variations of the WCB occurrence frequency. Both aspect are now discussed in the revised manuscript (l. 280ff).

**Grammar edits.**

L70 missing space "intrusions (Silverman..)"

We corrected the manuscript accordingly.

L370 remove "aims to" --> "UNet CNN that identifies"

We modified the text accordingly.